# Development characteristics of the rock fracture field in strata overlying a mined coal seam group

Yun Qi[1‡*], Wei Wang[1‡*], Jiaqi Ge[2*], Zebin Yang[3], Qingjie Qi[4]

**1** School of Coal Engineering, Shanxi Datong University, Datong, Shanxi, China, **2** Tashan Coal Mine, Jinneng Holding Coal Industry Group, Datong, Shanxi, China, **3** Changzhi Power Supply Company of State Grid Shanxi Electric Power Company, Changzhi, Shanxi, China, **4** China Coal Technology Engineering Group, Emergency Research Institute, BeiJing, China

‡ YQ and WW are co-first authors.
* qiyun_sx@sxdtdx.edu.cn (YQ); wangwei@sxdtdx.edu.cn (WW); g325815@163.com (JG)

**Data Availability Statement:** All relevant data are within the paper and its Supporting Information files.

**Funding:** Doctoral Research Startup Project of Shanxi Datong University (2020-B-18), the funder

## Abstract

The fracture development of the overlying strata after coal mining is an important guarantee of efficient gas drainage. In order to explore the fracture evolution characteristics close to a mined coal seam group, the $F_{15.16}$–24130 working face in the Pingdingshan No. 10 coal mine was taken as the research background. The FLAC3D numerical simulation software was used to study the migration and failure characteristics of the overlying strata during mining of a coal seam group, and the fracture evolution process of the stope was investigated. The results show that as the advancing distance increased, the fracture density and fracture height increased continuously due to deformation and failure of the overlying rock. The displacement of the overlying rock initially increased and then decreased, and the displacement of the floor rock initially decreased and then increased. When working face $F_{15.16}$ of the coal seam advanced to 75 m, a saddle-shaped plastic zone gradually formed in the upper part of the goaf and the floor of the goaf was formed. The pressure relief depth was proportional to the advancement distance. As the advancement distance of the working face increased, the pressure relief depth gradually extended to the $F_{17}$ coal seam, which was conducive to the development and penetration of the fractures in the coal floor and rock mass and was convenient for pressure relief gas drainage from the $F_{17}$ coal seam.

## 1 Introduction

During underground coal mining, the fracturing and movement of the overlying strata will evolve as mining progresses, and a large number of fractures will be generated in the overlying strata. Fractures are the channels for gas diffusion and flow, and their distribution characteristics have a significant impact on gas flow and extraction [1–3]. Pressure relief gas mainly accumulates in the fracture enrichment area above the goaf. The three zones of the overburden failure and mining fractures have a great impact on the safety of the coal mining [4, 5]. Therefore, studying the development characteristics of the fractures in the strata overlying the coal seam group plays an important role in gas control in the goaf and the upper corner and determination of the gas enrichment area.

is Yun Qi; Doctoral Research Startup Project of Shanxi Datong University (2020-B-08), the funder is Wei Wang. YES - Specify the role(s) played.

**Competing interests:** The authors have declared that no competing interests exist.

At present, many scholars have conducted a great deal of research on the distribution characteristics of the fractures and gas channels in the overlying strata after coal seam mining. For example, Liu Tianquan [6] and Qian Minggao et al. [7] developed the theory of three horizontal zones and three hard zones by studying the failure of the overburden and the distribution of the water conducting fracture zone. Yuan Liang [8] proposed the concept of a roof annular fracture circle and reported that there are vertical fracture development areas on both sides of the goaf. Based on rock fracture damage theory, Shi Biming et al. [9] used the RFPA software to explore the deformation characteristics and gas emission characteristics of the overlying coal seam during mining of the protective layer, and they reported that the failure area in the vertical direction of the overlying rock with the advancement of the working face is M-shaped. Li Shuqing et al. [10] explored the variation characteristics of the stress field during mining of the protective layer using the numerical simulation method, and they obtained the pressure relief protection criterion for mining of the protective layer. Yuan Dongsheng et al. [11] used the rock mechanics method and FLAC numerical simulation method to calculate the plastic zone around a rectangular tunneling roadway with a nearby protective layer and developed a method to reasonably determine the safety coal pillar. Xu Youping et al. [12] used the RFPA2D flow software to simulate and analyze the fracture distribution characteristics during fracturing, and revealed the fracture initiation, expansion, and extension processes in the coal body during directional hydraulic fracturing. Based on fractal theory and mathematical statistics, Li Hongyan et al. [13] qualitatively analyzed the distribution characteristics of the fracture field and divided the overburden fracture field in order to provide a theoretical basis for determining the appropriate gas area. Zhang Yong et al. [14] divided the fracture distribution range in front of the working face and the goaf based on mechanical theory and studied the conductivity characteristics of the gas migration channel. Based on the butterfly plastic zone theory of the surrounding rock, Ma Nianjie et al. [15] studied the high-quality gas channel and its development method in coal and gas CO mining, which greatly improved the gas drainage efficiency. Liu Hongyong et al. [16] established a theoretical model of the temporal and spatial morphology of the dominant gas channel in a fully mechanized mining face based on the theory of the mining fracture elliptical throwing zone, and they explored the relationship between the dominant gas channel and the advancement speed. The above research results play an important role in the distribution characteristics of the fractures in the overlying strata and gas prevention. However, these studies mainly focused on the distribution characteristics of the fractures and gas channels in the strata overlying a single coal seam being mined. However, for mining of a coal seam group, especially a close coal seam group, the formation of a fracture field and gas channels in the overlying strata is more complex than that for the mining of a single coal seam.

In view of this, in this study, the evolution of the fracture field and the formation of gas channels during the superimposed mining of a coal seam group was investigated using FLAC3D numerical simulations based on the research background of the $F_{15.16}$–24130 working face in the Pingmei No. 10 coal mine under the mining conditions of a close coal seam group. The results of this study provide a scientific basis for formulating and optimizing the drainage scheme of pressure relief gas and the safe and efficient drainage of pressure relief gas.

## 2 Development of the numerical model and parameter settings

### 2.1 Numerical simulation prototype

Taking the fully mechanized top coal caving face $F_{15.16}$–24130 in the Pingmei No. 10 coal mine as the research object, the $F_{15.16}$ coal seam was the mined coal seam. The direct roof is sandy mudstone plus a thin layer of carbonaceous mudstone and coal seam $F_{14}$, which is

relatively hard, with a thickness of 11–18 m. The main roof is grayish white fine to medium grained sandstone with a thickness of 10.9 m. The thickness of coal seam $F_{15}$ is about 1.9 m, the thickness of coal seam $F_{16}$ is about 1.5 m, the thickness of composite layer $F_{15.16}$ is 2.8–4.2 m, generally about 3.5 m, and the dip angle of the coal seam is 6–13˚, with an average of 9.5˚. The firmness coefficient of coal seam $F_{15}$ is f = 2–3. The thickness of the gangue between $F_{15}$ and $F_{16}$ is 0.2–0.7 m, and it gradually becomes thinner from outside to inside. The direct bottom is sandy mudstone with a thickness of 7.2–8.5 m. It is below coal seam $F_{17}$. The old bottom is sandy mudstone with a thin layer of fine sandstone and limestone, and the thickness is >10 m.

## 2.2 Mechanical model

The Mohr-Coulomb model was used for the numerical simulation. In the Mohr Coulomb model [17, 18], in the principal stress space of $\sigma_1$, $\sigma_2$, and $\sigma_3$, the corresponding strain components are the principal strains $\varepsilon_1$, $\varepsilon_2$, and $\varepsilon_3$. The three principal stresses and the corresponding strain increments obey the following relationship:

$$\sigma_1 \geq \sigma_2 \geq \sigma_3, \tag{1}$$

$$\Delta\varepsilon_i = \Delta\varepsilon_i^e + \Delta\varepsilon_i^p, \ (i = 1, 2, 3), \tag{2}$$

where e and P are the elastic and plastic parts, respectively, and the plastic component is only not 0 in the plastic flow stage.

According to Hooke's law, the strain increments and stress increments satisfy the following relationship:

$$\Delta\sigma_1 = \alpha_1 \Delta\varepsilon_1^e + \alpha_2 (\Delta\varepsilon_2^e + \Delta\varepsilon_3^e), \tag{3}$$

$$\Delta\sigma_2 = \alpha_1 \Delta\varepsilon_2^e + \alpha_2 (\Delta\varepsilon_1^e + \Delta\varepsilon_3^e), \tag{4}$$

$$\Delta\sigma_3 = \alpha_1 \Delta\varepsilon_3^e + \alpha_2 (\Delta\varepsilon_1^e + \Delta\varepsilon_3^e), \tag{5}$$

where $\alpha_1 = K + \frac{4G}{3}$, $\alpha_1 = K - \frac{4G}{3}$; $K$ is the bulk modulus; and $G$ is the shear modulus.

The principle of the failure criterion on the $(\sigma_1, \sigma_2)$ plane is shown in Fig 1.

In failure envelope $f(\sigma_1, \sigma_3) = 0$, from A to B, it is defined by the Mohr-Coulomb criterion $f^s = 0$:

$$f^s = \sigma_1 - \sigma_3 N_\phi + 2c\sqrt{N_\phi}, \tag{6}$$

$$N_\phi = \frac{1 + \sin(\phi)}{1 - \sin(\phi)}. \tag{7}$$

From B to C, it is defined by the tensile failure criterion $f_t = 0$,

$$f_t = \sigma_3 - \sigma^t, \tag{8}$$

where $\phi$ is the friction angle (˚), $c$ is the cohesion, and $\sigma^t$ is the tensile strength.

As can be seen from Fig 1, the value of $\sigma_3$ corresponding to the intersection of $f^s = 0$ and $\sigma_1 = \sigma_3$ is the ultimate tensile strength of the material. Therefore, the maximum tensile strength is

$$\sigma_{max}^t = \frac{c}{\tan\phi}. \tag{9}$$

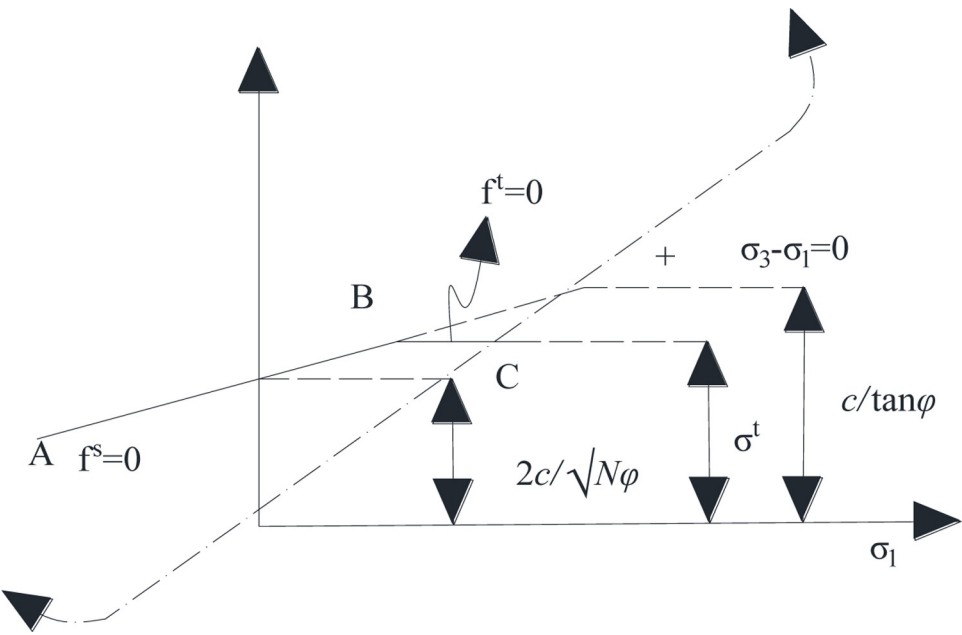

**Fig 1. FLAC3D Mohr-Coulomb criterion.**

## 2.3 Physical parameters and boundary conditions of the model

Generally speaking, in most cases, coal and rock masses are regarded as elastic-plastic media in numerical simulations. When the load on the coal and rock mass is less than its yield strength, it is linear elastic; and when the load is greater than its yield strength, it is plastic. The volumetric strain of the coal and rock mass is obvious when plastic deformation occurs, so the influence of the volumetric stress must be considered in the numerical calculations. Therefore, the elastic-plastic constitutive model was used in the simulation, and the Mohr-Coulomb criterion was used as the yield criterion. The relevant physical parameters of the coal and rock involved in the numerical calculations are presented in Table 1 [19, 20].

**Table 1. Physical parameters of the coal and rock.**

| Material Science | thickness (m) | density (kg·m$^{-3}$) | tensile strength (MPa) | cohesion (MPa) | bulk modulus (GPa) | shear modulus (Pa) | internal friction angle (°) |
|---|---|---|---|---|---|---|---|
| Fine to medium grained sandstone | >18 | 2600 | 6.5 | 13.4 | 57 | 3.40E+10 | 36 |
| Fine to medium grained sandstone | 1.8 | 2600 | 6.5 | 13.4 | 57 | 3.40E+10 | 36 |
| Coal seam $F_{14}$ | 0.4 | 2600 | 6.5 | 13.4 | 57 | 3.40E+10 | 36 |
| Sandy mudstone | 13 | 2400 | 4.1 | 6.8 | 46 | 2.42E+10 | 46 |
| Coal seam $F_{15.16}$ | 3.5 | 1410 | 1.8 | 2.1 | 25 | 1.48E+10 | 25 |
| Sandy carbonaceous mudstone | 2.8 | 2400 | 4.1 | 6.8 | 46 | 2.42E+10 | 46 |
| Coal seam $F_{17}$ | 2.5 | 1410 | 1.8 | 2.1 | 25 | 1.48E+10 | 25 |
| Gray packsand | 6 | 2600 | 7.8 | 15.4 | 62 | 3.74E+10 | 45 |
| Sandy mudstone | 4 | 2400 | 4.1 | 6.8 | 46 | 2.42E+10 | 46 |
| Limestone L1 | 4 | 2400 | 2.8 | 3.7 | 38 | 2.33E+10 | 28 |
| Coal line | 0.3 | 2400 | 2.8 | 3.7 | 38 | 2.33E+10 | 28 |
| Sandy mudstone | >10 | 2400 | 4.1 | 6.8 | 46 | 2.42E+10 | 46 |

**Table 2. Geostress parameters.**

| Maximum principal stress | | | Intermediate principal stress | | | Minimum principal stress | | |
|---|---|---|---|---|---|---|---|---|
| size (MPa) | direction (˚) | angle (˚) | size (MPa) | direction (˚) | angle (˚) | size (MPa) | direction (˚) | angle (˚) |
| 34.32 | −157.6 | −16.9 | 22.19 | −141 | 71.4 | 18.3 | −66.6 | −4.8 |

1. Displacement boundary condition: In the numerical analysis of the excavation influence area of coal seam $F_{15.16}$−24130 in the Pingmei No. 10 coal mine, the coal seam inclination of the working face was taken as the Y-axis, the coal seam strike was taken as the X-axis, and the Z-axis was along the direction of gravity. It was specified that the displacement of two symmetrical planes is 0, that is, when X = 0 and X = 787 are the constraint planes, the Y and Z directions are free; and when Y = 0 and Y = 150 are the constraint planes, the X and Z directions are free. Z = 0 is the constraint surface, which is constrained in the Z direction and free in other directions.

2. Stress boundary conditions: Based on the in-situ stress test results for the Pingmei No. 10 coal mine, the in-situ stresses of the measuring points near the research object are shown in Table 2.

Based on the above parameters and condition settings, combined with the data for the $F 2_{15.16}$−24130 fully mechanized top coal caving face in the Pingmei No. 10 coal mine, the coal strata with similar mechanical properties or small thicknesses were combined, and the three-dimensional model was established and meshed (Fig 2). All of the element types in the model were divided by 8-node hexahedral elements.

In order to record the variations in the stress and displacement of the mutual interference between the coal seams during mining of the working face, four monitoring lines were arranged in the model roof along the strike of the working face (Fig 3). The layout of the monitoring lines and the measurement point parameters are shown in Table 3. The four observation

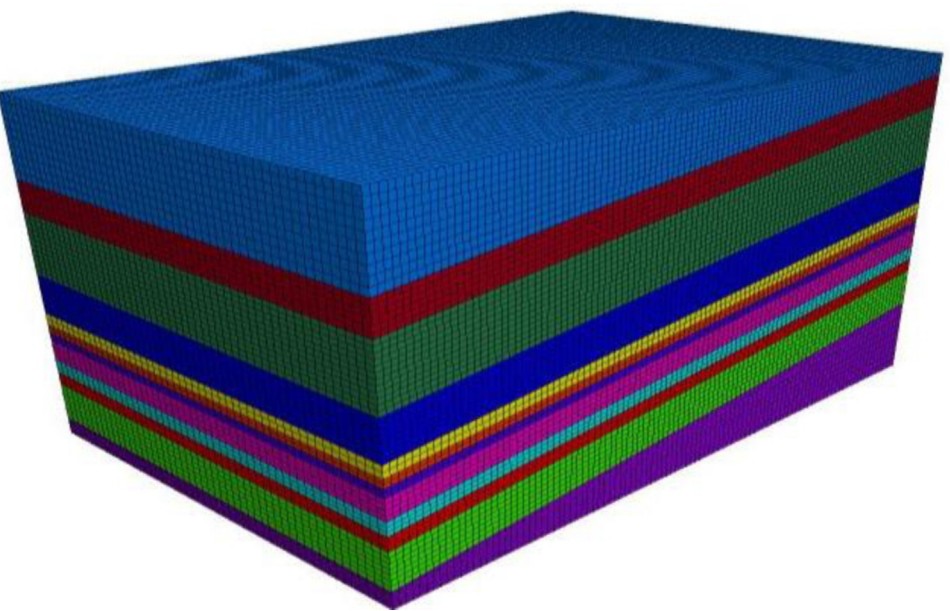

**Fig 2. Gridding graph of calculation model.**

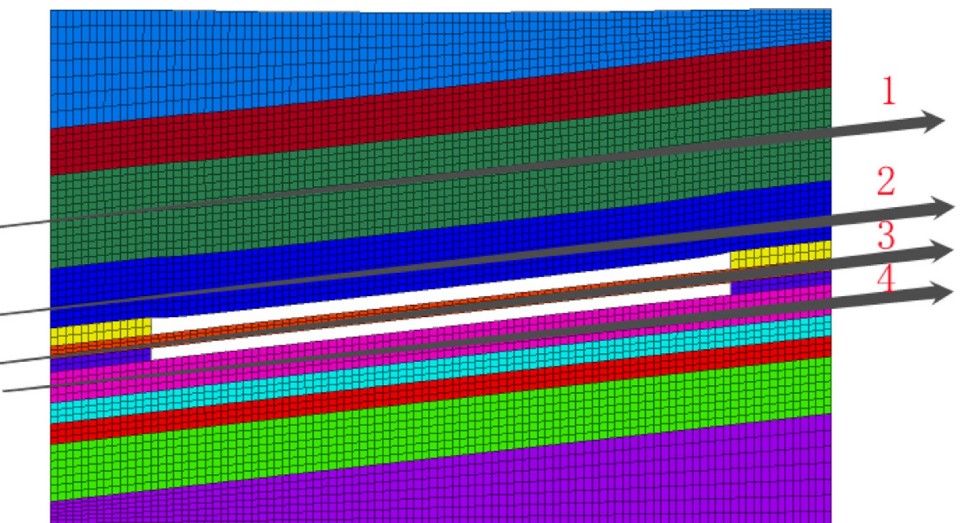

**Fig 3. Layout of monitoring lines.**

lines were set in the roof of coal seam $F_{15.16}$ (two observation lines), the middle rock stratum of coal seams $F_{15.16}$ and $F_{17}$ (one observation line), and in the floor of coal seam $F_{17}$ (one observation line). All of the observation lines were arranged along the center. During the mining process, the variations in the rock stress and displacement on each observation line were extracted.

## 3 Analysis of stress distribution characteristics of overburden in stope

In order to study the distribution of the surrounding rock stress at different advancement distances of the coal face, the working face of coal seam $F_{15.16}$ was excavated in 25 m intervals from the open cut, with a total excavation of 100 m, and then, coal seam $F_{17}$ was excavated using the same excavation method. When the model calculations converged, the excavation was completed. The working face excavation model was sliced along the coal seam's strike in the vertical direction, and the cloud diagram of the surrounding rock stress distribution during excavation is shown in Figs 4 and 5.

From the perspective of the change in the overburden stress, the overburden strata that were originally in a state of stress balance after the coal and rock mass excavation will be affected by the mining failure and its self weight, the stress will be redistributed, and stress concentration and pressure relief areas will develop around the excavation space, which will exhibit a dynamic change trend with the advancement of the working face [21]. It can be seen

**Table 3. Layout of monitoring lines and parameters of measuring points.**

| Serial number of observation lines | Rock properties at the observation lines | Horizon property | Measuring points | Distance from coal seam floor (m) |
|---|---|---|---|---|
| 1 | Fine to medium grained sandstone | Key layer | 100 | 22.5 |
| 2 | Fine to medium grained sandstone | Coal seam roof | 100 | 13.2 |
| 3 | Sandy carbonaceous mudstone | Interlayer between coal seams $F_{15.16}$ and $F_{17}$ | 100 | 3.9 |
| 4 | Gray packsand | Coal seam floor | 100 | 0 |

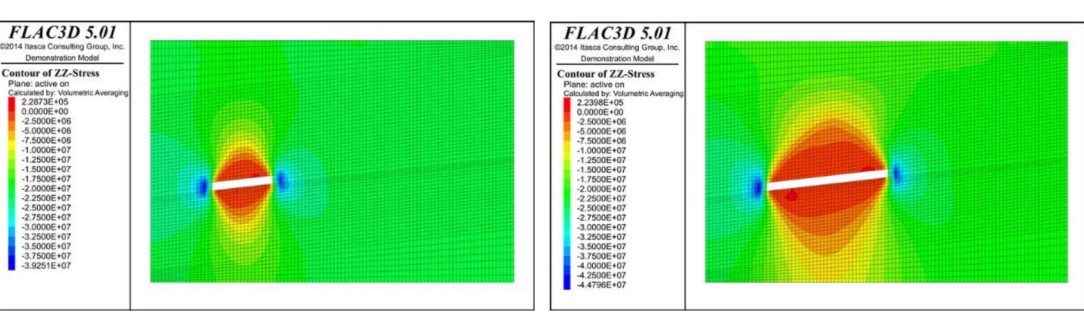

(a) advancement distance of coal seam $F_{15.16}$ is 25 m        (b) advancement distance of coal seam $F_{15.16}$ is 50 m

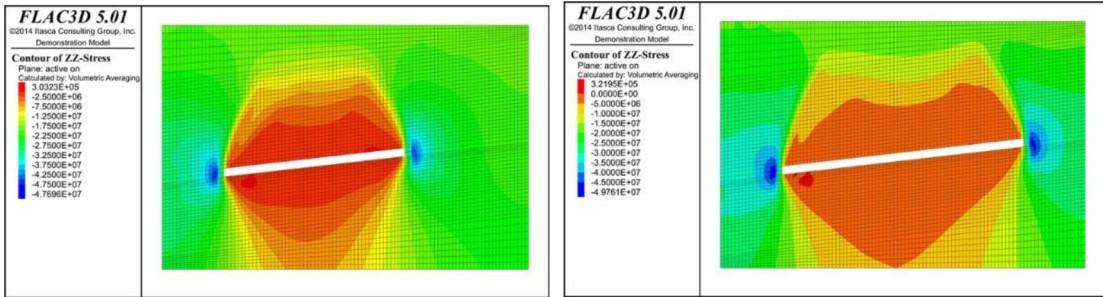

(c) advancement distance of coal seam $F_{15.16}$ is 75 m        (d) advancement distance of coal seam $F_{15.16}$ is 100 m

**Fig 4. Stress distribution in rock surrounding coal seam $F_{15.16}$ during excavation.**

from the stress change nephogram for the excavation of coal seam $F_{15.16}$ (Fig 4) that with the advancement of the working face, a blue stress concentration area is formed on both ends of the open cut, a red pressure relief area is formed in the overlying rock in the middle of the working face, and the stress concentration and pressure relief areas gradually grow and move forward continuously with the advancement of the working face. When the pressure value is negative, the coal and rock mass is subjected to compressive stress; and when the pressure value is positive, the coal and rock mass is subjected to tensile stress. In the vertical direction, the pressure relief range of the upper overburden in the goaf gradually increases with increasing advancement of the working face. In terms of its shape, the pressure relief area in the upper goaf gradually changes from a hump-like distribution in the early stage of coal mining to a saddle-shaped distribution as the advancement distance increases, and the pressure relief of the overlying strata close to the roof becomes more serious. The lower pressure relief zone is shaped like an inverted triangle, and its area increases with increasing advancement distance. In terms of stress, the compressive stress in the upper pressure relief area gradually increases from $-3.93\times10^7$ Pa to $-4.98\times10^7$ Pa, and the tensile stress in the stress concentration area gradually increases from $2.27\times10^5$ Pa to $3.22\times10^5$ Pa. It can be seen from (Fig 4A–4D) that the advancement of the working face leads to an increase in the pressure relief failure range of the upper part of the coal stratum. When the advancement reaches 50 m, the pressure relief height of the upper part tends to be stable, but the pressure relief depth of the lower part increases in proportion to the advancement distance.

As can be seen from Fig 5, the mining position of coal seam $F_{17}$ is located within the pressure relief protection range after coal seam $F_{15.16}$ is mined. The vertical stress of the surrounding rock during the mining of coal seam $F_{17}$ is small, and the pressure relief area in the upper

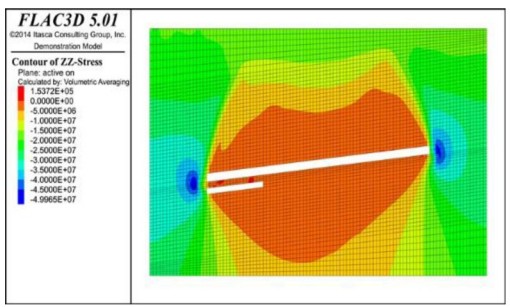
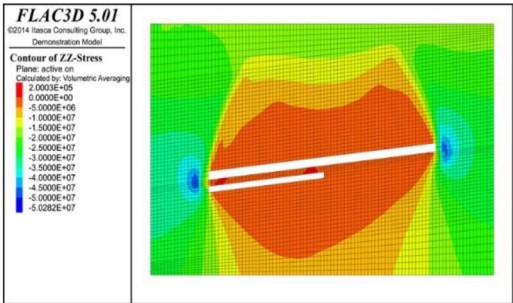

(a) advancement distance of coal seam $F_{17}$ is 25 m

(b) advancement distance of coal seam $F_{17}$ is 50 m

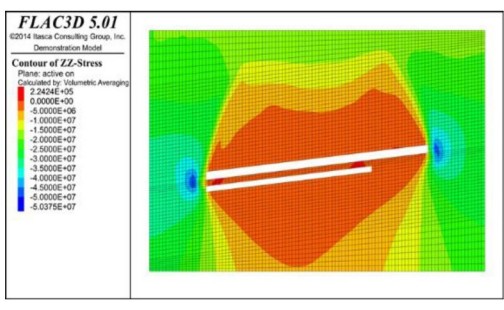
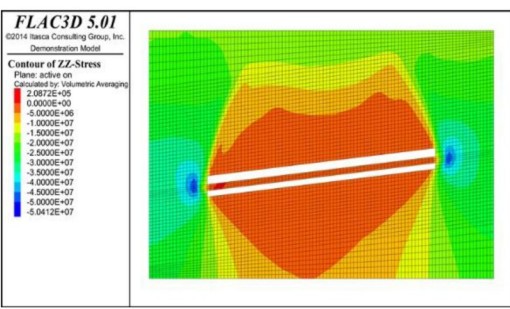

(c) advancement distance of coal seam $F_{17}$ is 75 m

(d) advancement distance of coal seam $F_{17}$ is 100 m

**Fig 5. Stress distribution in rock surrounding coal seam $F_{17}$ during excavation.**

part of coal seam $F_{15.16}$ remains saddle-shaped, but the compressive stress in the pressure relief area increases when it advances the same distance as that of coal seam $F_{15.16}$. When mining coal seam $F_{17}$, the compressive stress in the upper pressure relief zone gradually increases from $-5.00 \times 10^7$ Pa in the early stage of mining coal seam $F_{17}$ to $-5.04 \times 10^7$ Pa after mining. The tensile stress in the stress concentration area gradually increases from the maximum value of $1.54 \times 10^5$ Pa in the initial mining stage of coal seam $F_{17}$ to $2.24 \times 10^5$ Pa. The mining of coal seam $F_{17}$ does not have a large impact on the stress field, but it does increase the pressure relief depth in the lower part of coal seam $F_{17}$, while the pressure relief height of the coal layer overlying coal seam $F_{15.16}$ remains almost unchanged.

A can be seen from the stress change curves shown in Figs 6 and 7, the distribution of the stress change curve of the roof and floor during coal seam mining is basically consistent with the saddle-shaped stress nephogram. In the initial stage of the mining of coal seam $F_{15.16}$, the floor of coal seam $F_{17}$ has not been affected, and the floor stress does not change. With the continuous advancement of the working face of coal seam $F_{15.16}$, the stress change of the top and bottom slate of the coal seam tends to be stable, only the influence range increases with increasing advancement distance, and the influence of the mining of coal seam $F_{17}$ on the stress change law is not obvious.

## 4 Analysis of plastic failure characteristics of overburden

The variation in the overburden failure zone of the working face can be analyzed by studying the variation characteristics of the plastic failure zone [22, 23]. The development of the plastic failure zone in the strike direction of the coal and rock mass under different advancement distances is shown in Figs 8 and 9. In the simulated cloud diagram, none represents the area where no failure has occurred in the coal and rock mass, shear-n represents the area where

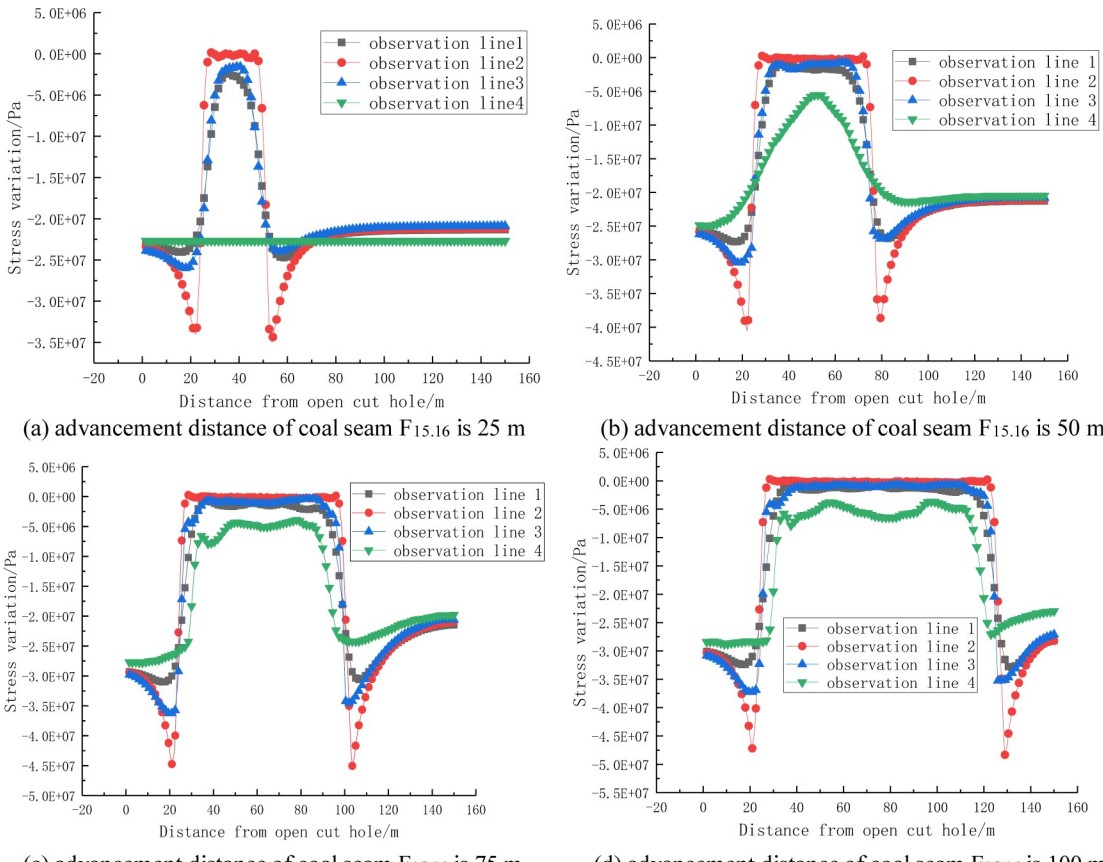

(a) advancement distance of coal seam $F_{15.16}$ is 25 m

(b) advancement distance of coal seam $F_{15.16}$ is 50 m

(c) advancement distance of coal seam $F_{15.16}$ is 75 m

(d) advancement distance of coal seam $F_{15.16}$ is 100 m

**Fig 6. Stress change curve of the rock surrounding coal seam $F_{15.16}$.**

shear failure is occurring in the coal and rock mass, shear-p represents the area where shear failure has occurred in the coal and rock mass, tension-n represents the area where tensile failure is occurring in the coal and rock mass, and tension-p represents the area where tensile failure has occurred in the coal and rock mass. It can be seen from the simulated cloud map that the distribution range of the overburden plastic zone in the working face is positively correlated with the advancement distance. The greater the advancement distance, the greater the distribution range of the plastic zone. From the analysis of the spatial distribution of the plastic failure of the overburden in goaf, in the initial stage of the mining of coal seam $F_{15.16}$, the overburden coal stratum initially undergoes shear failure under the influence of the mining and self-weight, cracks are preliminarily developed in the overburden coal stratum, the coal and rock mass above the working face is damaged within a small range under the influence of the shear stress, and the failure area is semicircular in shape. The advancement of the working face continues, and the coal and rock mass above the goaf begins to undergo tensile failure. In addition, the floor of the working face forms a floor plastic zone under the compressive stress of the coal and rock mass on both sides, and the range of this zone increases with increasing advancement of the working face. When the depth of the floor plastic zone reaches a certain level, it will no longer develop. However, as the working face advances, the development of the cracks in the floor of coal seam $F_{15.16}$ is conducive to the pressure relief gas drainage of coal seam $F_{17}$.

As is shown in Fig 8, as the working face advances, the overlying coal and rock mass in the goaf is damaged by the shear and tensile stress. Under the support of the coal pillar at the edge

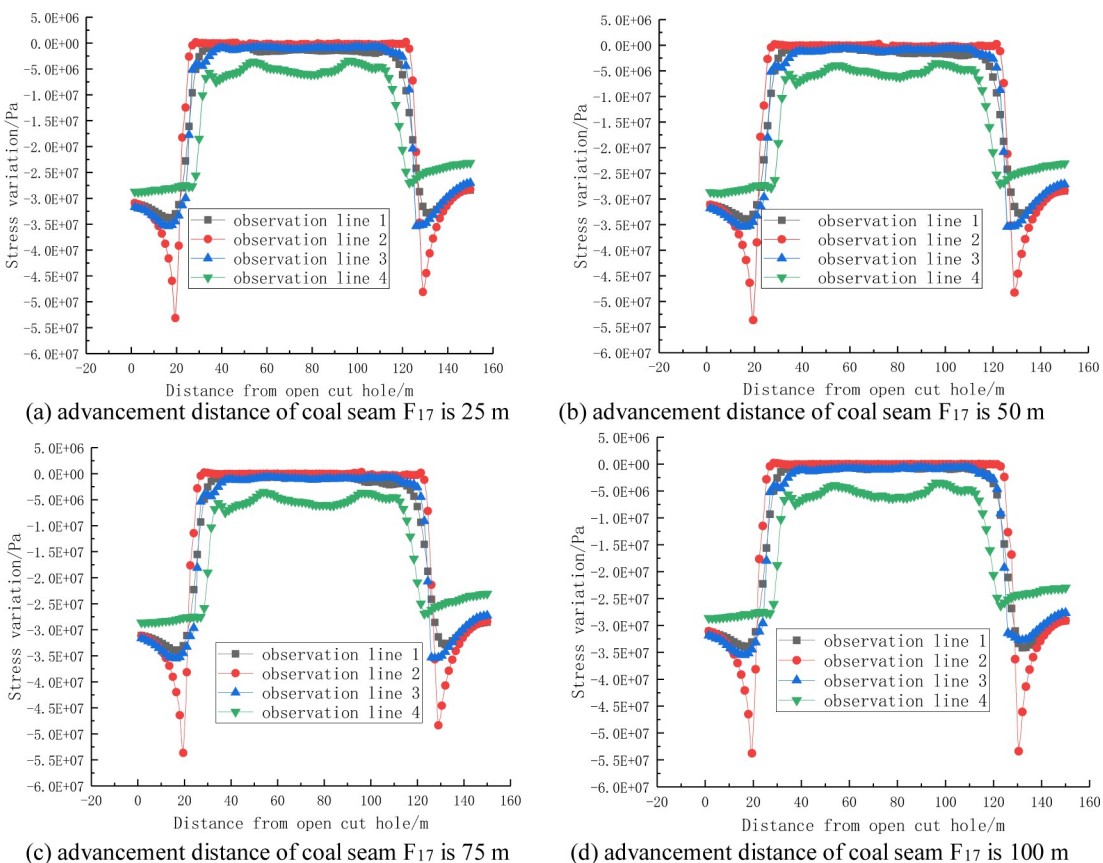

(a) advancement distance of coal seam $F_{17}$ is 25 m

(b) advancement distance of coal seam $F_{17}$ is 50 m

(c) advancement distance of coal seam $F_{17}$ is 75 m

(d) advancement distance of coal seam $F_{17}$ is 100 m

**Fig 7. Stress change curve of the rock surrounding coal seam $F_{17}$.**

of the goaf, the plastic deformation zone forms a saddle-shaped distribution. At this time, the overburden of the coal seam produces five areas from top to bottom: the local tension area, tension failure area, tension fracture area, and plastic deformation area. Among them, the coal and rock strata in the local tension area and tension failure area undergo large deformation due to tensile fracture and tear under the action of the tensile stress, forming a region dominated by collapse. The fractures in the tension fracture area are also fully developed, and the stress of the coal and rock strata in the upper plastic deformation area exceeds its yield strength, resulting in damage. The height of the development of the failure state of the coal and rock strata in this area is defined as the upper limit of the fracture zone. The stress deformation of the coal and rock strata in the elastic zone is small, so this is the bending subsidence zone. As is shown in Fig 9, the mining of coal seam $F_{17}$ has little impact on the distribution of the overburden plastic zone of coal seam $F_{15.16}$, and the change in the working face advancement distance along the strike direction of the working face has little impact on the plastic zone of the bottom plate, which is almost unchanged after developing to a certain depth in the bottom plate. Along the vertical direction, the change in the distribution of the plastic zone in the roof of the working face is not obvious, and the development height of the plastic zone basically tends to be stable. It can be seen that due to the pressure relief of the working face of coal seam $F_{15.16}$, the coal and rock mass stress of the roof and floor is released. During the mining of coal seam $F_{17}$, the development range of the floor plastic zone is small and there is almost no secondary development.

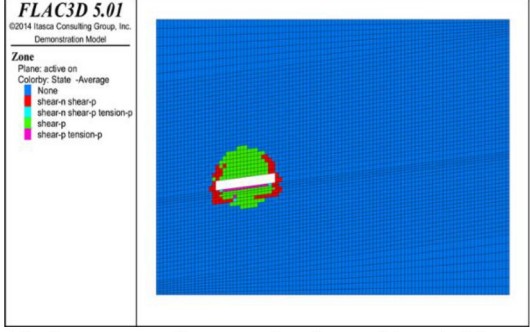
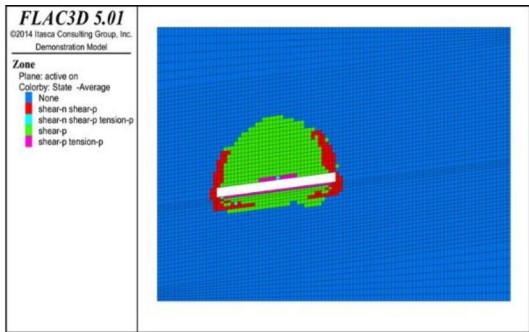

(a) advancement distance of coal seam $F_{15.16}$ is 25 m (b) advancement distance of coal seam $F_{15.16}$ is 50 m

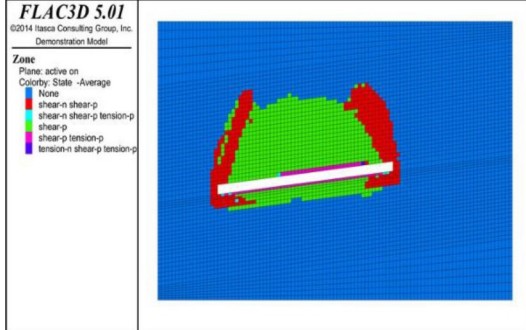
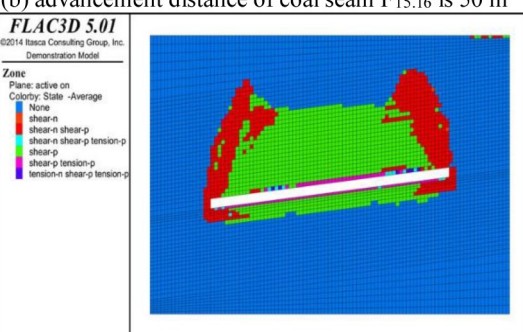

(c) advancement distance of coal seam $F_{15.16}$ is 75 m (d) advancement distance of coal seam $F_{15.16}$ is 100 m

**Fig 8. Distribution map of plastic failure in coal seam $F_{15.16}$.**

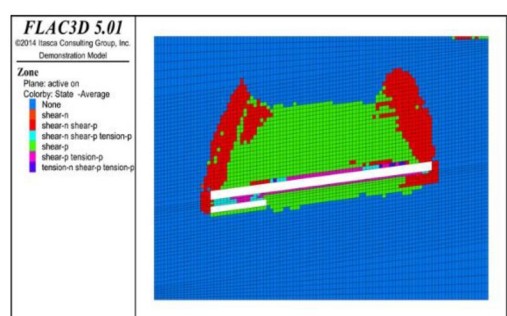
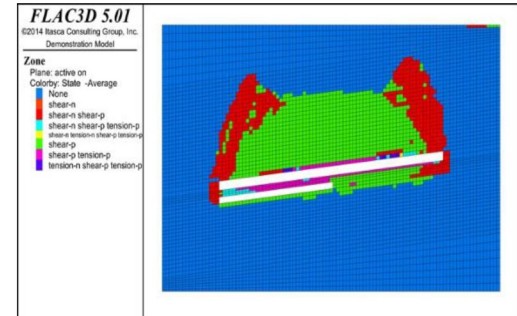

(a) advancement distance of coal seam $F_{17}$ is 25 m (b) advancement distance of coal seam $F_{17}$ is 50 m

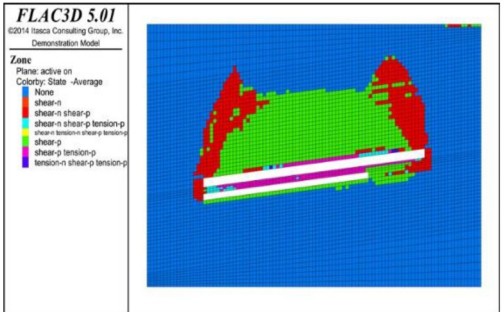
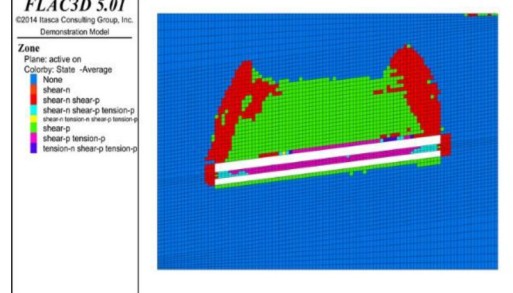

(c) advancement distance of coal seam $F_{17}$ is 75 m (d) advancement distance of coal seam $F_{17}$ is 100 m

**Fig 9. Distribution map of plastic failure in coal seam $F_{17}$.**

## 5 Analysis of variations in overburden displacement

With the advancement of the working face, the coal and rock mass of the roof and floor of the coal seam gradually deforms until plastic failure occurs under the influence of the mining, and finally, a caving zone, fracture zone, and bending subsidence zone form in the vertical direction above the goaf. The fracture development caused by the coal and rock deformation during the advancement of the working face can be expressed by its shape variable [24, 25]. The variation nephogram of the overburden displacement along the strike direction at working face advancement distances of 25 m, 50 m, 75 m, and 100 m is shown in Figs 10 and 11. It can be seen from Fig 10 that the roof and floor coal strata of coal seam working face f15.16 exhibit obvious displacement changes as the advancement distance of the working face increases, and the roof and floor coal and rock mainly exhibit downward and upward vertical movement, respectively. As the advancing distance of the working face increases, the displacement of the roof coal and rock body exhibits a linear growth trend, and the displacement of the floor coal and rock layer also exhibit a small upward displacement with increasing advancement of the working face, which increases from the initial upward displacement of 8.6 cm to the final displacement of 11.69 cm. It can be seen that as the working face advances, plastic failure occurs in the overlying rock and underlying rock. The top plate bends and sinks, and the bottom plate bulges. It can be seen from Fig 11 that under the influence of the mining of the protective layer of coal seam $F_{15.16}$, the coal and rock masses at the top and bottom of coal seam $F_{17}$ undergo a small change in displacement relative to coal seam $F_{15.16}$, and its displacement with respect to the rock stratum between coal seam $F_{15.16}$ gradually changes to a downward displacement. As the advancement distance of the working face increases, the roof of coal seam $F_{17}$ begins to exhibit downward vertical displacement, and the coal and rock mass of the floor exhibits upward displacement.

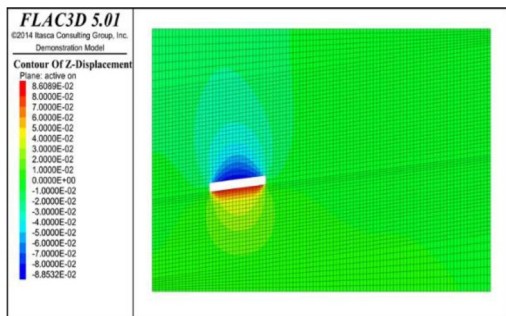

(a) advancement distance of coal seam $F_{15.16}$ is 25 m

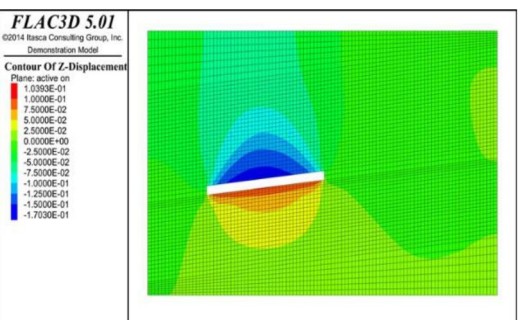

(b) advancement distance of coal seam $F_{15.16}$ is 50 m

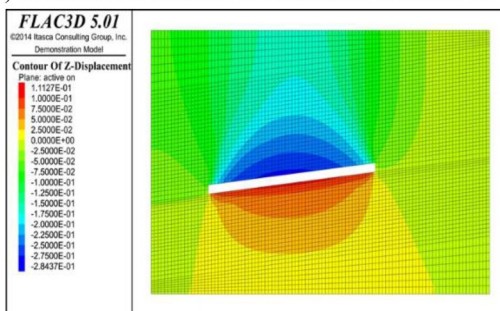

(c) advancement distance of coal seam $F_{15.16}$ is 75 m

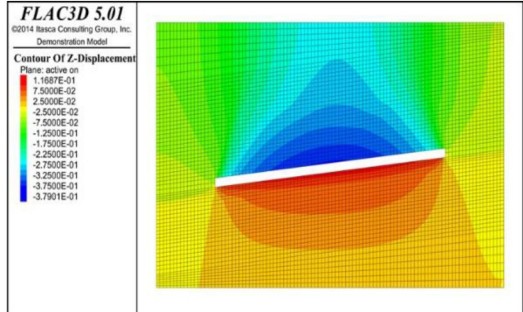

(d) advancement distance of coal seam $F_{15.16}$ is 100 m

**Fig 10. Displacement of rock surrounding coal seam $F_{15.16}$.**

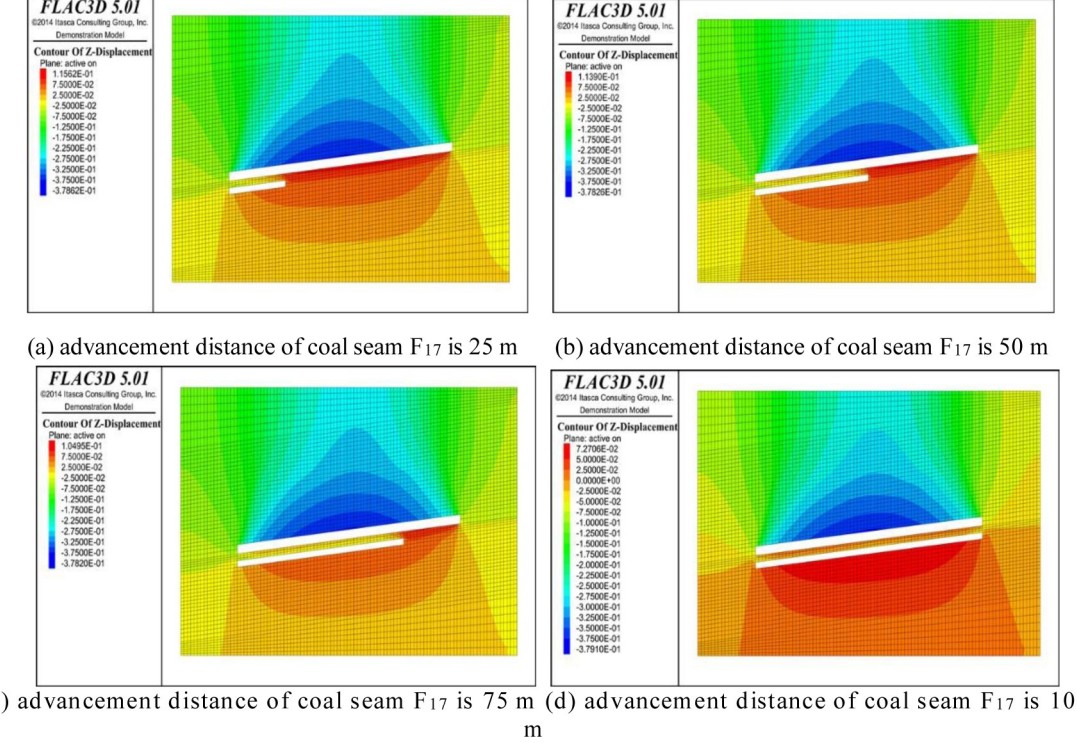

(a) advancement distance of coal seam $F_{17}$ is 25 m

(b) advancement distance of coal seam $F_{17}$ is 50 m

(c) advancement distance of coal seam $F_{17}$ is 75 m

(d) advancement distance of coal seam $F_{17}$ is 100 m

**Fig 11. Displacement of rock surrounding coal seam $F_{17}$.**

By comparing Figs 10 and 11, it was found that under the influence of the dual mining of coal seams $F_{15.16}$ and $F_{17}$, the displacement of the overlying strata initially increases and then decreases, increases, and decreases, which is consistent with the conclusion obtained from the physical similarity tests of the overlying strata fracture morphology due to mining and its evolution conducted by Li Shugang et al. [26], that is, the fracture development occurs in the overlying coal strata under the influence of mining of the single coal seam. After the changes in the expansion and compaction due to repeated mining, the fracture development undergoes re-expansion and re-compaction.

It can be seen from Figs 12 and 13 that the change in the displacement of the rock stratum is inversely proportional to its distance from the coal seam being mined, and the maximum change in the displacement of the same rock stratum occurs in the middle of the goaf. In addition, as the advancement distance increases, the area of the change in displacement of the coal and rock mass affected by the mining also increases, and the displacement change curve of the roof rock stratum forms an $\Omega$-shaped distribution, i.e., high in the middle and low at both ends, The variation curve of the floor heave forms an inverted $\Omega$-shaped distribution, i.e., low in the middle and high at both ends. Generally speaking, the mining of coal seam $F_{17}$ has little effect on the change in the displacement of the roof and bottom slate of coal seam $F_{15.16}$.

## 6 Conclusions

1. The rock strata at the top and bottom of the working face undergo pressure relief failure. The overlying rock of the working face undergoes elastic deformation as the advancement distance increases, and it gradually collapses or fractures, resulting in a large number of

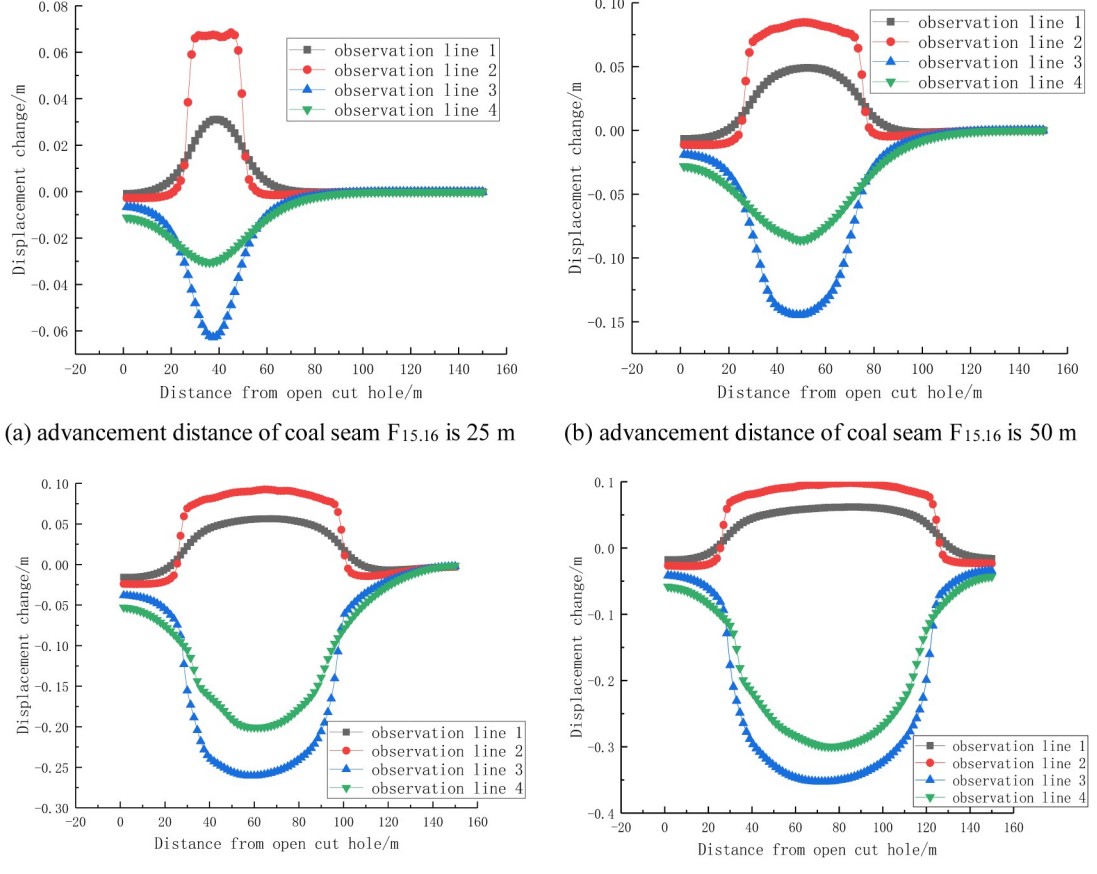

(a) advancement distance of coal seam $F_{15.16}$ is 25 m

(b) advancement distance of coal seam $F_{15.16}$ is 50 m

(c) advancement distance of coal seam $F_{15.16}$ is 75 m

(d) advancement distance of coal seam $F_{15.16}$ is 100 m

**Fig 12. Change in displacement for different advancement distances of coal seam $F_{15.16}$.**

fractures. The floor of the working face also also has floor heave due to stress, resulting in a large number of cracks.

2. As the advancement distance of the working face increases, the fracture density and fracture development height continuously the first increase and then decrease. This change is reflected by the change in the displacement, that is, the change in the displacement of the overlying rock initially increases and then decreases, and the change in the displacement of the floor rock initially decreases and then increases.

3. Supported by the coal pillars on both sides, the fractures in the coal and rock mass are fully developed under the action of the tensile stress. When the working face of coal seam $F_{15.16}$ is advanced to 75 m, a saddle-shaped plastic zone gradually forms in the upper part of the goaf, and the pressure relief height tends to be stable at this time. The goaf floor forms an inverted triangular plastic zone, and the pressure relief depth is directly proportional to the advancement distance.

4. As the advancement distance of the working face increases, the pressure relief depth gradually extends to coal seam $F_{17}$, which is conducive to the development and connection of the fractures in the coal and rock mass in the floor and the pressure relief due to the extraction of coal seam $F_{17}$.

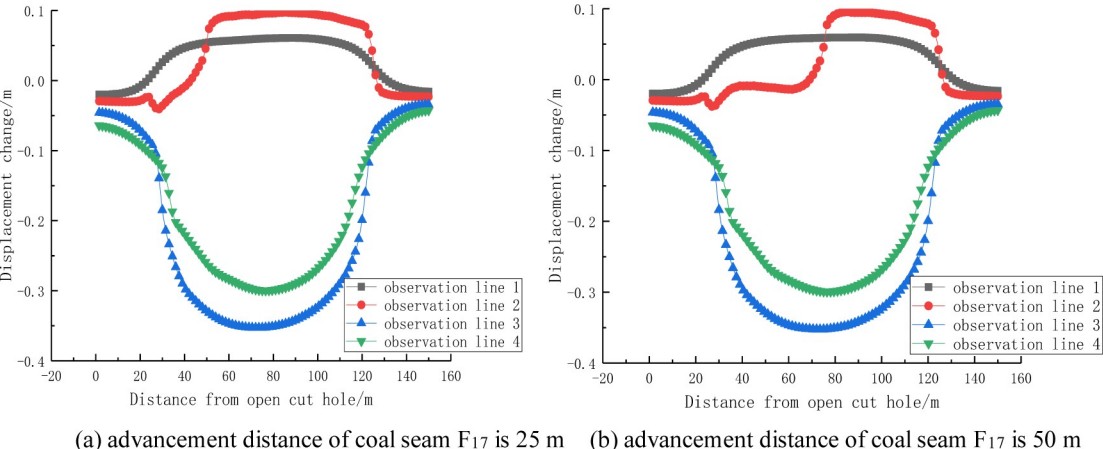

(a) advancement distance of coal seam $F_{17}$ is 25 m   (b) advancement distance of coal seam $F_{17}$ is 50 m

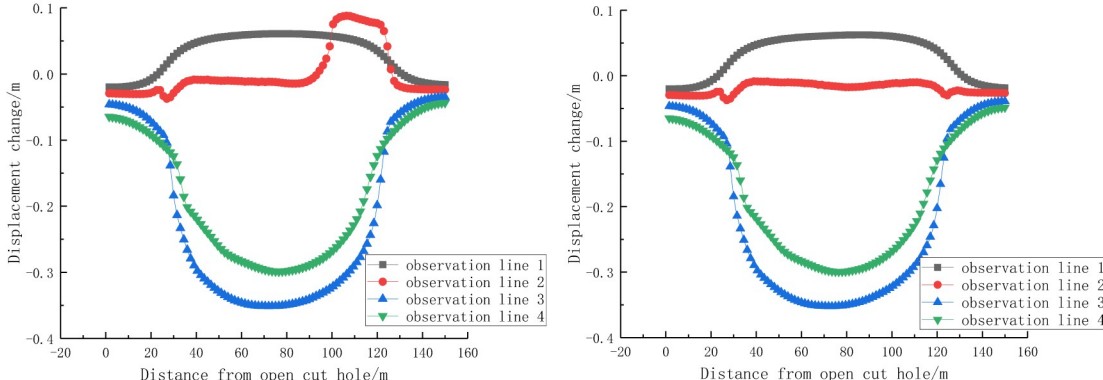

(c) advancement distance of coal seam $F_{17}$ is 75 m (d) advancement distance of coal seam $F_{17}$ is 100 m

**Fig 13. Change in displacement for different advancement distances of coal seam $F_{17}$.**

## Supporting information

**S1 File.**
(DOCX)

**S1 Data.**
(XLSX)

## Acknowledgments

We thank LetPub (www.letpub.com) for its linguistic assistance during the preparation of this manuscript.

## Author Contributions

**Data curation:** Yun Qi, Wei Wang.

**Formal analysis:** Zebin Yang, Qingjie Qi.

**Funding acquisition:** Yun Qi, Wei Wang.

**Resources:** Jiaqi Ge, Qingjie Qi.

**Supervision:** Zebin Yang, Qingjie Qi.

**Writing – original draft:** Yun Qi, Wei Wang.

**Writing – review & editing:** Yun Qi, Wei Wang, Jiaqi Ge.

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
