## [Decision Letter · Decision Letter 0]

12 Apr 2022

PONE-D-22-00440Development Characteristics of the Rock Fracture Field in Strata Overlying a Mined Coal Seam GroupPLOS ONE

Dear Dr. Qi,

Thank you for submitting your manuscript to PLOS ONE. After careful consideration, we feel that it has merit but does not fully meet PLOS ONE’s publication criteria as it currently stands. Therefore, we invite you to submit a revised version of the manuscript that addresses the points raised during the review process.Please do your best to address the comments raised by reviewer No. 2

We look forward to receiving your revised manuscript.

Kind regards,

Khalil Abdelrazek Khalil, Ph.D.

Academic Editor

PLOS ONE

Journal Requirements:

2. ‘Please include your tables as part of your main manuscript and remove the individual files. Please note that supplementary tables (should remain/ be uploaded) as separate "supporting information" files.’

Additional Editor Comments (if provided):

Please address the comments raised by reviewer 2

Reviewers' comments:

Reviewer's Responses to Questions

**Comments to the Author**

1. Is the manuscript technically sound, and do the data support the conclusions?

Reviewer #1: Partly

Reviewer #2: Yes

2. Has the statistical analysis been performed appropriately and rigorously? 

Reviewer #1: Yes

Reviewer #2: Yes

3. Have the authors made all data underlying the findings in their manuscript fully available?

Reviewer #1: Yes

Reviewer #2: Yes

4. Is the manuscript presented in an intelligible fashion and written in standard English?

Reviewer #1: Yes

Reviewer #2: Yes

5. Review Comments to the Author

Reviewer #1: The article is innovative and can be published.The English level of the article meets the requirements.The number of references is relatively small, and it can be appropriately increased to about 25 references.

Reviewer #2: 1. The conclusion should be concise and rewritten.

2. The letters or parameters in Figure 1 should be clear.

3. For numerical simulation, why did not the effect of fault structure on rock movement be considered?

4. The basis for the values of various moduli in Table 1 should be explained in detail.

6. PLOS authors have the option to publish the peer review history of their article (what does this mean?). If published, this will include your full peer review and any attached files.

Reviewer #1: No

Reviewer #2: No

---

## [Author Response · Author response to Decision Letter 0]

25 Apr 2022

Dear editor and reviewers:

Thank you for your letter and the reviewers’ comments on our manuscript entitled " Development Characteristics of the Rock Fracture Field in Strata Overlying a Mined Coal Seam Group". Those comments are very helpful for revising and improving our paper, as well as the important guiding significance to other research. We have studied the comments carefully and made corrections which we hope meet with approval. The main corrections are in the manuscript and the responds to the reviewers’ comments are as follows (the replies are highlighted in blue ).

I checked the list of all references, and I can guarantee the integrity and correctness of each reference.

Replies to the reviewers’ comments:

Response to reviewers 

Reviewer #1: The article is innovative and can be published.The English level of the article meets the requirements.The number of references is relatively small, and it can be appropriately increased to about 25 references. 

Response: References have been added as required. At present, there are 26 references, and the newly added references have been marked in blue font.

Response to Reviewer #2: 

1. The conclusion should be concise and rewritten.

Response: The conclusion part has been condensed again with concise language

2. The letters or parameters in Figure 1 should be clear.

Response:Figure 1 has been modified as CAD source drawing, and all its letters and parameters are clearly visible

3. For numerical simulation, why did not the effect of fault structure on rock movement be considered?

Response:Because the mechanical strength of the rock stratum at the fault zone is lower than that of the surrounding rocks, and the fractures are relatively easy to develop, the evolution law of rock fractures at the fault zone can not truly reflect the evolution law of overburden fractures affected by mining under the mining conditions of coal seams. In addition, there is no fault structure in the working face within the research scope of the project group, However, the influence of fault structure on the evolution of overburden fractures will be considered in the subsequent research process.

4. The basis for the values of various moduli in Table 1 should be explained in detail.

Response: References 20 and 21 have been added to the paper. These two references are the project reports of the research group. The acquisition process of various data in Table 1 is described in these two references.

---

## [Decision Letter · Decision Letter 1]

12 May 2022

Development Characteristics of the Rock Fracture Field in Strata Overlying a Mined Coal Seam Group

PONE-D-22-00440R1

Dear Dr. Qi,

We’re pleased to inform you that your manuscript has been judged scientifically suitable for publication and will be formally accepted for publication once it meets all outstanding technical requirements.

Kind regards,

Khalil Abdelrazek Khalil, Ph.D.

Academic Editor

PLOS ONE

Additional Editor Comments (optional):

Reviewers' comments:

Reviewer's Responses to Questions

**Comments to the Author**

1. If the authors have adequately addressed your comments raised in a previous round of review and you feel that this manuscript is now acceptable for publication, you may indicate that here to bypass the “Comments to the Author” section, enter your conflict of interest statement in the “Confidential to Editor” section, and submit your "Accept" recommendation.

Reviewer #2: All comments have been addressed

2. Is the manuscript technically sound, and do the data support the conclusions?

Reviewer #2: Yes

3. Has the statistical analysis been performed appropriately and rigorously? 

Reviewer #2: Yes

4. Have the authors made all data underlying the findings in their manuscript fully available?

Reviewer #2: Yes

5. Is the manuscript presented in an intelligible fashion and written in standard English?

Reviewer #2: Yes

6. Review Comments to the Author

Reviewer #2: The author has made detailed modifications and can be published in the current form. I look forward to receiving the news of the publication of this manuscript.

7. PLOS authors have the option to publish the peer review history of their article (what does this mean?). If published, this will include your full peer review and any attached files.

Reviewer #2: No

---

## [Editor Report · Acceptance letter]

26 Sep 2022

PONE-D-22-00440R1 

Development Characteristics of the Rock Fracture Field in Strata Overlying a Mined Coal Seam Group 

Dear Dr. Qi:

I'm pleased to inform you that your manuscript has been deemed suitable for publication in PLOS ONE. Congratulations! Your manuscript is now with our production department. 

Kind regards, 

on behalf of

Dr. Khalil Abdelrazek Khalil 

Academic Editor

PLOS ONE